# Relocated Employees' Experience with the Costs and Benefits of Video Technology for Maintaining Relationships

**Kayla Walling**

Department of Behavioral Sciences, Liberty University,1971 University Blvd, Lynchburg, VA 24515, USA;
kmwindsor@liberty.edu

**Abstract:** Relocation as a result of a job opportunity or as part of a current job has increased in recent years in the United States. The frequency with which employees are relocating makes exploring employees' perception of the costs and benefits of technology for maintaining family relationships useful. The purpose of this qualitative descriptive study was to explore how employees who have relocated for employment opportunities perceive the costs and benefits of video technology as an option for maintaining family relationships, for employees working in a company located in the United States. The sample consisted of 25 employees at a healthcare facility in the state of Florida who had relocated for employment opportunities and had used video technology to maintain relationships with family members in another location. Data collected from semi-structured interviews, open-ended questionnaires, and a focus group were analyzed through thematic analysis. Findings provided evidence that relocated employees perceived video technology to be an important tool to help maintain work–life balance when they were away from their family. The employees' perception of the benefits of video technology in maintaining family relationships included closeness despite distance and tangibility of the person. These findings have implications for employees who have relocated, employees considering relocation, and families of employees, as well as organization leaders and human resource departments.

**Keywords:** video technology; family relationships; employee relocation





## 1. Introduction

Relocation as a result of a job opportunity or as part of a current job has increased in recent years in the United States (Weng et al. 2018). Career professionals may hold different opinions of relocation, and likewise make career decisions differently based on such factors as the length of time they have been a member of the workforce (Patton and Doherty 2020). Many factors can impact an employee's decision regarding whether relocating for a job will be a worthwhile endeavor. One of the most influential factors that can impact an employees' willingness to relocate is their family (Patton and Doherty 2020). While institutions may incentivize career relocation by means of incentives, these incentives do not address the sensitive solutions families must come up with to make relocation possible, such as finding new methods to stay in contact and forging new communal bonds (Patton and Doherty 2020).

### 1.1. Employee Relocation and Family

A large body of literature focuses on employee relocation and factors that impact willingness to relocate (Chapa and Wang 2016; Garwood 2014; Ullrich et al. 2015). Relocation entails moving to a new locality, whether it is a new county, state, or even country. Many factors can impact an employee's decision as to whether relocating for a job will be a worthwhile endeavor. Ullrich et al. (2015) noted spousal support as a major factor. If a spouse is willing to relocate with their partner who is relocating for work, they are more likely to accept the opportunity and be willing to deal with leaving their current community and support system (Ullrich et al. 2015). This finding was also echoed in Eby

and Russell (2000), who found spouses' attitudes to be significantly stronger predictors than employee attitudes and perceptions and background factors. Similarly, Patton and Doherty (2020) noted that the government often incentivizes career relocation by means of tax incentives, housing benefits, and other rewards, but the family factor is not addressed by these incentives, and families must come up with solutions to make relocation possible, such as finding new methods to stay in contact and forging new communal bonds (Patton and Doherty 2020).

*1.2. Technology for Video Communication*

Video technology, such as webcams and Skype, were introduced in the mid-1990s, allowing users to visually interact with each other despite being separated by geographical barriers (Dhaliwal 2021). This technology, which allows users to participate in a phone call with the additional element of bi-directional live video footage, has clear and direct applications for both professional and consumer use (Jana et al. 2016). Video chatting has been used to improve social contingency and facilitate learning in toddlers (Myers et al. 2017). Physicians have used video chatting to distribute high-quality ultrasound images in developing countries (Levine et al. 2015). Families and friends have used video chatting to participate in geocaching, a form of "treasure hunting" undertaken with the help of GPS coordinates, over long distances (Procyk et al. 2014). Meetings held over video chat have been implemented as a component of an adolescent weight loss intervention in order to improve program outcomes (Ptomey et al. 2015). Customer service has been made more efficient with some service or retail providers offering instant video chat to get fast answers to simple inquiries (Parise et al. 2016).

Among the aspects of video chat that differentiate it from other communication technologies upon analysis, Sindoni (2014) noted eye contact, technological integration of speech, sight, and text capabilities, and the fact that various chat resources mimic face-to-face conversation unlike any other mode of communication (Sindoni 2014). Services offered over video chat may improve the accessibility of certain services that are usually only accessible to those living within commuting distance of a physical location (Melton et al. 2015). However, Zhou and Feng (2017) noted that whether or not individuals answered video calls they received was directly related to how much they anticipated they would enjoy the call if it was leisure-based, and how useful they thought the call would be when it was work-based.

*1.3. Benefits and Costs*

The benefits of video chat as a communication technology primarily center on the extensive number of purposes and needs it can successfully serve, especially in comparison to other forms of technology-mediated communication such as phone calls. An example of video chat technology solving a problem identified in the existing literature is the sharing of emotional once-in-a-lifetime experiences in real time (Inkpen et al. 2013; Massimi and Neustaedter 2014). With the development of mobile video chat, people can watch important events live, ask questions, comment, and more (Massimi and Neustaedter 2014). Video chat may allow for interpersonal bonds to be forged and maintained in a more emotionally connected way than through other forms of electronic communication (Sherman et al. 2013). Sherman et al. (2013) found that after in-person bonding, video chat communication may be the most beneficial means for improving or strengthening emotional bonds. Teenagers are likely to see the primary benefits of video chatting as staying in touch with friends they may not be able to see otherwise and facilitating group activities, such as study sessions, remotely (Buhler et al. 2013). Business professionals likely see the primary benefits of video chatting as staying in touch with family and friends, as well as participating in remote meetings and seminars for work (Jana et al. 2016).

Using video chat as a communication technology also has potential costs. Existing literature reveals that the majority of the costs which are associated with this technology center around technological issues and application design flaws that should be corrected

(Jana et al. 2016). Another significant cost of video chatting is that it depends directly on the quality of mobile networks which are accessible to the users: if the communication infrastructure in place cannot handle data speeds above a certain rate, the video chat experience will be mediocre at best (Oduor et al. 2014).

### 1.4. Technology and Communication with Family

In recent decades, technology has become increasingly prevalent and sometimes necessary for people to effectively stay in contact with family members; thus, researchers have explored the use of communication technologies within families (Stein et al. 2016). Oduor et al. (2014) found that the participants primarily used communication technologies to interact with family about their well-being, life advice, economic support, and general everyday activities. Oduor et al. (2014) also noted how varied technological infrastructure, which was dependent on where family members lived, affected the accessibility of certain technologies for different participants. Some family members, particularly those who are elderly, may not be inclined to use communication technologies for any purpose besides staying in contact with younger family members (Tsai et al. 2016).

Some researchers have recently examined the use of video chat technologies among family members, although the relatively recent development of video chatting means that this body of literature is less than extensive (McClure and Barr 2017). McClure and Barr (2017) highlighted the applicability of video chatting to both experiences of family book reading and parenting interventions, showing how video chat communication may enrich family bonds when family members are living apart due to career relocation or other circumstances. A major benefit of video chatting among family members is the unique opportunity to share visual and audible family experiences across long distances (Inkpen et al. 2013). With the use of video chat, experiences can be shared in real time, as opposed to recorded and then shared as a file after an event has already ended. Cramer and Jacobs (2015) specifically explored how cohabitating couples chose to communicate daily. With the multitude of options available, using different modes of communication can add importance and meaning, and emphasize urgency (Cramer and Jacobs 2015). However, Massimi and Neustaedter (2014) noted that while video chat technology is uniquely useful for sharing events in real time that would otherwise need to be recorded and shared, certain technical challenges associated with existing video chat technology called for improving certain design elements of video chat applications. Additionally, the question of if and when video chatting an event is appropriate is also important, such as if it is okay to "Skype-in" to a funeral or other serious event (Massimi and Neustaedter 2014).

### 1.5. Current Study

As organizations are continuing to increase globally, employment relocation has become more frequent (Weng et al. 2018). While relocating can be a positive experience for some employees, others may not enjoy relocation, possibly leading these employees to be driven out of either their field or current job to avoid frequent relocation (Patton and Doherty 2020). Moreover, one of the most influential factors that employees take into consideration during relocation is their ability to maintain relationships with their family (Patton and Doherty 2020). If employees know how video technology can maintain family relationships, this knowledge may make the decision to relocate less stressful in regard to the aspect of family relationships.

The frequency with which employees are relocating makes exploring employees' perception of the costs and benefits of technology for maintaining family relationships useful. In an increasingly globalized world, studies addressing work–life balance have focused on the role of technology in bridging the gap between employment and family life (Gan 2021; Kędra 2020; Martín-Bylund and Stenliden 2020). Despite the limitations of technology in terms of face-to-face interactions, this innovation has helped many employees to meaningfully continue interacting with their families (Dhaliwal 2021). In the existing literature, technology has been primarily examined in terms of work efficiency and effectiveness

(Attaran et al. 2019; Brougham and Haar 2018; Cakula and Pratt 2021), but not in terms of the private family life of employees. The changing use of sophisticated technology as a means for communicating with family generates a need to study video technology and family relationships (Kędra 2020; Martín-Bylund and Stenliden 2020). However, it was not known how employees who have relocated for employment opportunities perceive the costs and benefits of video technology as an option for maintaining family relationships. This study was an exploration of employees' perception of the costs and benefits of using video technology for maintaining family relationships of employees who have relocated for employment opportunities. The aim was to provide information that can be used by employers who seek to transfer employees to other geographic locations and by employees who are considering either relocation or have other interest in changing their work–life balance.

This study extended prior research on employment-based relocation (Patton and Doherty 2020; Weng et al. 2018) and technology by addressing the gap between the employees' perception of the costs and benefits of using video technology as an option to maintain family relationships during employment-based relocations. Two theories were used to form the theoretical foundation of the study and guide the research questions: social exchange theory (Stafford and Kuiper 2021) and attachment theory (Johnson 2019). Based on social exchange theory, benefits are rewards that add value to a person's relationship, whereas costs are factors that have negative value to a person's relationship (Stafford and Kuiper 2021). Social exchange theory was utilized as a framework appropriate for an individual's perception of the cost and benefit in a social situation. Attachment theory suggests that people relate relationships to attachment figures and control how close or distant they allow themselves to get to others (Johnson 2019). Utilizing social exchange theory and attachment theory as a framework, this study was a qualitative exploration of how employees who have relocated for employment opportunities perceive the costs and benefits of video technology in the maintenance of familial relationships.

The purpose of this qualitative descriptive study was to explore how employees who have relocated for employment opportunities perceive the costs and benefits of video technology as an option for maintaining family relationships for employees working in a company located in the United States. The phenomenon that was explored in the study was employees' perception of the costs and benefits of video technology in maintaining family relationships for employees who have relocated for employment opportunities. The following main research question and sub-questions were used to guide the study:

R1. How do employees who have relocated for employment opportunities perceive the use of video technology in maintaining family relationships in the United States?

R1a. How do employees who have relocated for employment opportunities perceive the benefits of video technology in maintaining family relationships in the United States?

R1b. How do employees who have relocated for employment opportunities perceive the costs of video technology in maintaining family relationships in the United States?

## 2. Methodology

To conduct this study, a qualitative descriptive research design was used. Qualitative research was found to be appropriate for this study due to the focus on perceptions of individuals. Additionally, the descriptive research design was used as descriptive studies present a complete summary of an occasion in the average language common to that occasion (Siedlecki 2020). The geographic location selected was the United States, thus excluding employees who have relocated to another country. The general population included all employees who have relocated for employment opportunities within the United States and who have left their families in another location. The participants were selected purposively, which means that the selection of participants was not random but based on specified criteria (Campbell et al. 2020). The target population were all employees at a healthcare facility located in the state of Florida who (a) had relocated for employment,

(b) had left their families in another location, and (c) use video technology to maintain family relationships.

The sample included 25 employees at a healthcare facility in the state of Florida who had relocated for employment opportunities and used video technology to maintain family relationships with family members that were in another location. Among the 25 participants, 10 participated in the semi-structured interviews that lasted approximately 40 min, 10 participated in an open-ended questionnaire that took approximately 30 min, and 5 participated in the focus group discussion that lasted 43 min. For the semi-structured interviews and open-ended questionnaire, the rationale for the sample size was based on data saturation, a tentative point wherein no new information can be generated because all of the key elements of a phenomenon have already been identified (Hennink and Kaiser 2022). Hennink and Kaiser (2022) found that 9 participants was the minimum number of participants needed to reach data saturation in qualitative studies, which justifies the sample size for this study. For the focus group discussion, the decision to use 5 participants was based on the findings of Hennink and Kaiser (2022), who found that four was the minimum number for reaching saturation for focus groups.

In order to access the target population, site authorization from one healthcare company, whose headquarters are in Naples, Florida, was secured. The site authorization was secured by visiting the headquarters of this healthcare company that has multiple branches within the United States. The rationale for this decision was to increase the eligible pool of workers who might fit the criteria for participation in this study because of the possibility of being assigned in another branch. Seeking authorization from the leaders of said company entailed providing a document detailing a short background of the study, the assurance of confidentiality, the scope of participation of the potential participants, and the specific eligibility requirements to be part of the study.

After site authorization was formally secured from the leaders of the company and IRB approved the study, the recruitment of participants commenced. Emails were sent to all employees of the company detailing the criteria for participants. Contact details such as an email address were included so that interested individuals could directly communicate and further discuss any concerns or questions about the study. Screening occurred with participants through email to ensure that they met the eligibility criteria. Once they were vetted for inclusion in the sample, they were asked which data source they would like to participate in.

The research tools for conducting the current study included semi-structured interviews, an open-ended questionnaire, and a focus group discussion. The use of three sources was to attain triangulation of multiple sources of data, which is one of the main processes involved in the empirical investigation of a phenomenon using a qualitative methodology (Yin 2013). The questions in the semi-structured interviews were informed by the theoretical framework, which included social exchange theory (SET) and attachment theory. A series of open-ended questions were developed based on the tenets and concepts central to the two theories of the theoretical framework. These questions were intended to provide relevant information about video chat communication within families who are living apart due to job relocation. Sample questions were: What is your view regarding the use of video technology in maintaining family relationships during a work-related relocation? What are the benefits of using video technology in maintaining family relationships during a work-related relocation?

Another source of data for this study was open-ended questionnaires. The open-ended questionnaires were collected online through Survey Monkey. The open-ended questionnaires gathered data on employees who had just relocated for a job, improved work–life balance of the workforce, and more harmonious family relationships. These questions were also informed by the theoretical framework. Sample questions were: What is your view regarding the use of video technology in maintaining family relationships during a work-related relocation? What are the benefits of using video technology in maintaining family relationships during a work-related relocation?

A third source of data for the study was a focus group discussion. The focus group discussion involved employees who had relocated for employment opportunities within the United States. Participants were asked to participate in a Skype discussion online. Information on how employees who had relocated for employment opportunities in the United States perceived the costs and benefits of video technology as an option for maintaining family relationships was gathered through the focus group.

To ensure trustworthiness of the data, a number of measures were taken. Member checking was used to assess the validity of the interview and focus group transcripts. Negative cases and rival explanations were used to strengthen the study's credibility. Data triangulation was also used to strengthen the credibility of the findings. The study's transferability was enhanced through the generation of thick description. An audit trail was created to document the research procedure. The establishment of evidence was practiced in order to establish the study's dependability. Peer debriefing was also used to enhance the study's dependability. The explicit identification of the researcher's reflexivity was made in order to enhance the study's confirmability.

Ethical measures were taken to ensure the ethical soundness of the study. Informed consent was secured from the selected organization where the participants were recruited. Site authorization was secured by visiting the organization which has multiple branches within the United States. The approval of the Internal Review Board was secured before data collection began. By submitting the required documents and forms, all requirements of the IRB were fulfilled. Data security was maintained by assigning codes to conceal the identities of the participants and storing the data in password-protected files.

The individual semi-structured interviews were conducted online through Skype. The individual interviews were approximately 40 min. The interviews were audio recorded via Skype in preparation for the analysis in the later stage of the study. The open-ended questionnaires were collected online through Survey Monkey. The link to the questionnaire was provided through email. The questionnaire took approximately 30 min to complete. The focus group discussion was conducted through Skype. The focus group lasted for 43 min. The focus group was audio recorded in preparation for the analysis in the later stage of the study.

The raw data were prepared for analysis by transcribing the interviews and the focus group discussion and extracting the answers to the open-ended questionnaire. The transcription process was accomplished through a software called Rev. All transcripts were stored in a Microsoft Word document, properly documented with the assigned codes corresponding with the source of the data. All data from interviews, focus group, and open-ended questionnaires were extracted and compiled in NVivo. To answer the research question and the corresponding sub-questions, thematic analysis was used for the analysis of data collected from semi-structured interviews, the open-ended questionnaire, and focus group discussion. The specific coding strategy used is the method developed by Saldana (2016), utilizing first- and second-cycle coding. The first-cycle coding involved assigning labels to portions of text based on their purported essence, whereas second-cycle coding involved a more focused segmentation of text based on their salient meaning in relation to the research questions.

## 3. Results

### 3.1. Demographic Characteristics

Out of the 25 participants who took part in the study, 10 were young career professionals with age ranges from 31 to 40 years old. Further, 80% of the participants were White and 60% were married. Across the different data sources, 27% held management positions. The 15 participants of the semi-structure interviews and focus group reported using video technology an average of 1.3 times daily and 7.8 times weekly. Further details are presented in Table 1.

**Table 1.** Demographic Profile of Study Participants.

|  | Semi-Structured Interview | Focus Group |
| --- | --- | --- |
| Gender | | |
| Male | 6 | 1 |
| Female | 4 | 4 |
| Age range | | |
| 30 and below | 0 | 5 |
| 31–40 | 7 | 0 |
| 40–50 | 2 | 0 |
| 50 and above | 2 | 0 |
| Ethnicity | | |
| White | 8 | 4 |
| Native American | 1 | 0 |
| African American | 1 | 1 |
| Marital status | | |
| Single | 3 | 3 |
| Married | 7 | 2 |
| Average # of job relocations | 2.4 | 1.6 |
| Management position | 3 | 1 |
| Full-time employee | 10 | 5 |
| Part-time employee | 0 | 0 |
| Video chat technology averages | | |
| Times per day | 1.3 | 1.4 |
| Times per week | 7.9 | 7.6 |

Confidentiality of personal information was assured for all participants and pseudonyms were assigned to anonymize personal identity. The researcher used "PSI" to name the participants from semi-structured interviews, "POS" for online open-ended questionnaire participants, and "PFG" for participants who joined the focus group discussion. The researcher assigned a number for each participant joining the activities. For example, "PSI01" for Participant 1 who participated in a semi-structured interview.

*3.2. Perceived Intimacy Created through Video*

Thematic analysis of interview data in triangulation with open-ended questionnaires and a focus group informed the answers to the research questions and yielded themes that addressed the research questions. This main research question examined the general views of United States employees who relocated for employment purposes concerning the use of video technology in maintaining the family relationship. The researcher identified the major theme "perceived intimacy created through video" as a general concept that defines the perceived use and benefits of video technology for employees who work away from their family.

The participants across the three data sources described how video technology bridges the physical gap between the employee who works far away from home and their family and relatives. In different situations and marital circumstances, all participants expressed the value of staying in contact with the family. PFG01 felt that "the further away you are from your family, it [video technology] is even more beneficial".

PSI04 described how he maintained his romantic and platonic relationships with people he valued most using video technology. PSI10 who described himself as a dad of toddlers and wanted to closely watch the development milestones of the children perceived video technology to be beneficial in his work. PSI02 who worked in the West Coast area for nine years shared how video technology aided him in establishing the closeness he had with his nephews, siblings, and parents even though they lived in the east coast area of the United States. PSI02 shared, "It's allowed me to keep my gainful employment that I did not want to lose, and also be closer to family and be 3000 miles away". Using Skype or FaceTime, PSI02 "got to see things" at home while maintaining gainful employment.

Ten of the participants in the open-ended questionnaire also expressed that video technology made the separation of families easier as they got to see each other using various video technology applications. Two of the participants in the focus group expressed that video technology gave them the feeling of being close to their homes. All participants in the individual interviews, focus group, and open-ended questionnaires mentioned how they were able to visualize and feel their respective homes without spending time and money to travel back home. These views are evident in the responses of the participants in all six questions. For instance, PFG02 shared that video technology allowed her to see the faces at home and be able to share the daily activities after work without spending her money and time away from work.

From the pool of interview participants, two of these participants directly mentioned visualization of the person and home. The concept of cost-effectiveness in relation to visualization of home and the people at home was further elaborated in the focus group discussions with five participants.

### 3.3. Audio-Visual Communication and Emotional Security as Benefits

The second research question (R1a) for this study was: How do employees who have relocated for employment opportunities perceive the benefits of video technology in maintaining family relationships in the United States? Twenty-five employees participating in focus group discussions, semi-structured interviews, and open-ended questionnaires affirmed the benefits of video technology in maintaining family relationships when working away from home. These assertions from the participants were grouped to develop two themes: the interface allows for audio-visual communication and allows for emotional security for the users.

The capability of the technology to provide audio-visual interaction gave the users an immersive experience that simulated a seemingly real presence of a person within a similar environment. This immersive experience is evident in statements of participants such as "feel like home," "close to home," "see faces," and "hear and see", among others.

PSI01 described the enticing benefits video technology brought to communication. PSI01 considered video technology an "effective form of communication". Video technology for him entices children to communicate with him unlike any other communication system. Its effectiveness in communicating with children was also expressed in the experiences of PSI10. He mentioned that with the adult perspectives of accepting the transition of working away, relocation was not too difficult. However, relocation experiences with children are difficult. Like PSI01, relocation without video technology would make it difficult for him to accept job opportunities away from his family.

The positive contribution video technology offers to family relationships is also expressed in the experiences of PSI09. As an older employee, his relationship with his grandchildren is important. He shared that with video technology, "I've noticed that my grandchildren recognize me in person, even if I haven't seen them in months". The effectiveness of video technology in visual and audio communication was also expressed by PSI04. With video technology, PSI04 was able to participate in her father's birthday while simultaneously doing her work. PSI01 also stressed the effectiveness of visuals in communicating with her family. She implicated that the regular weekend FaceTime strengthened the family connection which they could only establish during holidays such

as Thanksgiving and Christmas. She observed that regular video communication helps in getting to know a person and helps in establishing relationships.

From the pool of online respondents, POS01 cited that the daily visual contact "helps maintain closeness with children/grandchildren". POS03 provided a quick explanation concerning the positive benefits of face-to-face interactions in maintaining family relationships. POS03 said: "virtual presence at family events, and more frequent real time interactions which brings more context to the experience compared to phone calls, texts, emails".

PFG04 established her connection with young nephews and nieces using video technology where she is able "to have face to face conversation with them". Although she confessed that video communication is not enough compared to physical presence, she considered even the 5 min conversation "precious and treasured". PFG04 said that with video technology, she is able to "talk about everyday life things" with her parents and siblings. PFG01 used the term "beneficial" to describe how video technology maintains the relationship an employee has with loved ones. PFG01 believed that with the power of technology, an employee is "able to stay in the community with your family and loved ones". PFG01 felt that being with the community alleviates the feeling of isolation.

Eight out of the twenty-five participants verbatim expressed the psychological benefit of video technology in terms of emotional security through seeing and feeling the community with family and friends despite being away for work. These views were evident in the participants' reminiscences of their homes and families and expressions of isolation. These views are described by the participants in each data source. PSI01 mentioned "stress," "homesickness," and "frustrations" to describe the psychological benefits video technology provides in maintaining the family relationship. PSI06 mentioned that seeing the "full picture" of the surroundings and her home is a benefit that video technology offers to someone employed away from home. PSI07 shared a similar thought with PSI06, saying that: "the main benefit would be just being able to keep up with each other to make sure everything is secure, and that you're doing okay". Video technology assures her that everything back home is well taken care of despite her absence.

Three of the open-ended questionnaire respondents shared the value of audio-visual interaction contributing to employees' sense of emotional security. Among these respondents is POS05 who believed that while there are people who "become too emotional when they see their home setting", others such as he considered this communication platform as "an important tool to maintain psychological health". POS05 described video technology as a tool that "allows one to maintain a very important visual connection to their loved one". POS08 also shared this view, using the statement "makes me feel close to home."

*3.4. Absence of Physical Connection and Technical Issues as Costs*

The third research question (R1b) for this study was: How do employees who have relocated for employment opportunities perceive the costs of video technology in maintaining family relationships in the United States? Twenty-five employees participating in focus group discussions, semi-structured interviews, and open-ended questionnaires affirmed that employees' perception of the major cost video technology intends to address among employees who were relocated for employment is the absence of physical connection. The participants also mentioned challenges that further let them feel the absence of physical aspects of a relationship. The perceived absence of physical connection in video technology is further exacerbated with major challenges. These challenges represent costs to employees who have relocated because these factors serve as barriers in maintaining relationships with one's family.

Across the data sources, the participants articulated the relevance of technology in connecting the family despite the geographical distance. This is evident in the answer of PSI02, who pointed out the benefits of video technology and how it superficially addresses the physical absence of a person from their loved ones. She emphasized that there are limitations to what a person relocated away from the family can do despite the great ability of video platforms to connect people. PFG04 also expressed that with the limited attention

span of children and aging parents, she could not get more call time with them. PFG04 said that while video technology tried to connect people, "it can sometimes make me want more . . . and I realized I can't". PFG03 described this experience as "false sense that is enough". She further implicated that physical connection is irreplaceable. PFG03 shared that she may be able to see and talk to her loved ones on a daily basis, but that these are not sufficient to eliminate her "longing to be with them". PFG04 added that while availability of video technology made the relocation decision-making process easier, the daily use of the platform made "me long to see them in person. Whenever I do see them in that face to face video it makes me long for the day where I can physically be with them".

Although all participants stressed that internet connections were widely accessible in their areas, network issues either with their families or with their locations posed challenges during video calls. Network connectivity was perceived as a cost because family members cannot communicate with each other when they lose internet and network connection. Six of the respondents from the open-ended questionnaire identified poor network service as a challenge in the use of video technology. POS08 shared: "Sometimes reception/connection can interfere while we have a long day ahead of us, we always have to put it off until our schedules match". Like POS09, POS02, and POS07 who briefly mentioned "internet connectivity", POS10 also said: "The few challenges we've run into are usually relating to lack of service or wi-fi. Personally, just finding time that works across time zones can be a challenge." POS05 further explained that relocation areas are sometimes rural with "poor internet connectivity".

While all participants did not indicate their technical competency in using video technology, their family members such as aging parents and children mostly depend on technologically inclined family members to be able to communicate with them. Technical capability was perceived as a cost because employees and their family members might find it difficult to communicate if one member does not know how to use technology.

PSI01 shared that older members of the family "are not as technologically savvy" and that learning the latest software is among the hurdles they need to surpass. PSI01 explained that as video technology and all associated software are new, there is "some angst about trying to use it". PSI01 shared that using video technology with a family who has younger-generation members is not as challenging. Two open-ended questionnaire respondents mentioned the technical capability of the users as an issue in video technology. POS04 stressed that "both parties have to be able to use video technology in order for it to work". Three participants mentioned that the older members of the family are less technology savvy and that they are the group for whom it is most difficult to participate in video communication. PFG05 described her grandparents' and mother's frustrations in using video technology. PFG04 also said: "The same thing like with my grandma, like she doesn't do video anything and so our conversations were very short on the phone". PFG05 said they ended up using phone calls because "older relatives, maybe it's harder to get them to do that". PFG02 further described technology in relation to the technical competency of the user as "great when it works, but when it doesn't work for whatever reason or if it's just a signal, it can be very, very frustrating".

## 4. Discussion

### 4.1. Discussion of Findings

The first theme discussed how employees who have relocated for employment opportunities perceived the benefits of video technology in maintaining family relationships in the United States. The participants experienced perceived intimacy created through the use of video technology. This finding is supported by different findings of previous studies about how employees value staying in contact and how video technology can make this possible (Patton and Doherty 2020). Multiple studies have shown that individuals primarily use communication technologies to interact with family about their well-being, life advice, economic support, and general everyday activities (Gan 2021; Kędra 2020; Martín-Bylund and Stenliden 2020). Similarly, the findings of the current study highlight the importance

of participants communicating with family members through video technology as a way to update loved ones about their well-being and day-to-day activities that might make it seem that they are close to each other despite the distance.

The participants in this study revealed that, through video technology, they can visualize the people they want to communicate with and that it is also cost-effective and time-efficient. The participants shared that video technology allows for emotional security because people feel more connected when communicating using video. These findings extend the current literature about the use of video technologies. Previous research has shown how families build relationships using video chat (Gan 2021; Kędra 2020; Martín-Bylund and Stenliden 2020). The topic of families building relationships using video chat, particularly those with young children who are living away from other family members, has been explored (Gan 2021; Kędra 2020; Martín-Bylund and Stenliden 2020), including how it allows for more authentic reactions and emotions to be shared surrounding such events, as emotions and thoughts about an event can be shared as they are cognitively formed instead of being recalled after the fact (Gan 2021; Kędra 2020; Martín-Bylund and Stenliden 2020).

The main cost of using video technology in maintaining family relationships in the United States was addressing the physical absence of a person from their loved ones. While there were benefits in using video technology, the participants still felt that there were limitations to how video technology can connect family members together. This finding is supported by Dhaliwal (2021), who mentioned that technological challenges may hinder the sharing of some family experiences. Video technology still cannot replace the physical connection that face-to-face encounters can provide.

The finding that network connectivity is a challenge of video technology is supported in the literature. Another significant drawback of video chatting is that is depends directly on the quality of mobile networks which are accessible to the users (Gan 2021; Gan et al. 2020). Moreover, if video chat applications are poorly designed or maintained, users can experience significant lag time, dropped calls, and other problems (Gan 2021; Gan et al. 2020). This can be particularly concerning when individuals or groups are using video chat to communicate across different types of localities, as rural areas tend to have less effective or efficient communication infrastructure in place.

*4.2. Recommendations for Practice*

Video technology has been found to be useful for employees who have relocated for employment opportunities because of perceived intimacy created through video and the fact that it allows audio-visual communication. This will benefit employees who have relocated and their families because they can communicate with their families, despite being geographically separated for long periods of time due to work. Employers can also benefit from this result as employees would not easily be discouraged from relocating because they could still communicate with their family through video technology.

Organizational leaders and human resource (HR) departments of organizations should consider the results of this study in improving policies in the organization regarding employees who have relocated. They should develop organizational policies that addresses the needs of employees who have relocated. Network connectivity is seen as a cost of using video technology. The organizations must assure the employee who has relocated that the internet connection in the new location will not be a problem. If it is possible, internet connection can be added as part of the compensation package for the employee who has relocated. Employees who have relocated should also have adequate support for video technology.

Technical capability was also found to be a cost of using video technology, especially for older members of a family. The employee and other members of the family should work together so that the elderly members of their family can still communicate with the employee who has relocated. They can either teach the elderly members how to use the



technology or be available to assist whenever older relatives want to call the employee who has relocated.

### 4.3. Recommendations for Future Research

The findings of this study revealed that employees who have relocated experienced perceived intimacy created through using video technology. Further research is needed to have a deeper understanding of how video technology creates perceived intimacy as a way to maintain family relationships. More studies are needed so that organizations can assist employees who relocated.

Future researchers should conduct more studies about how employees who have relocated for employment opportunities perceive the costs and benefits of video technology as an option for maintaining family relationships. There were several costs and benefits mentioned in the findings, such as video technology allowing emotional security for its users, but there is also the issue of absence of a physical connection. Further research is warranted to fully understand the role of technology, not in terms of work efficiency and effectiveness, but in terms of its role in the private family life of employees.

The current study is limited by the sample size and research design. Specifically, the low sample size and qualitative research design limits the generalizability of the results. There is a need to increase sample size of the study to increase the significance level of the results, in order for the results to be representative of the population for the phenomenon being examined. Quantitative methodology also allows for a large sample size and generalizations from the population.

Future researchers could also examine other contexts such as military deployment. The current study explored the experiences of employees who have relocated and how they maintain relationships with their family members. A study about how members of the military use video technology to maintain relationships with their family members could contribute to a deeper understanding of how video technology can be used in this context.

**Funding:** This research received no external funding.

**Institutional Review Board Statement:** The study was conducted in accordance with the Declaration of Helsinki, and approved by the Institutional Review Board of Grand Canyon University. IRB-2018-589.

**Informed Consent Statement:** Informed consent was obtained from all subjects involved in the study.

**Data Availability Statement:** Not applicable.

**Conflicts of Interest:** The author declares no conflict of interest.

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
