# Peer review of "Relocated Employees’ Experience with the Costs and Benefits of Video Technology for Maintaining Relationships"

_socsci, doi:10.3390/socsci12050286_

Round 1
Reviewer 1 Report
A brief summary:
The article entitled "Relocated Employees' Experience with the Costs and Benefits of Video Technology for Maintaining Relations" concerns a very important and current issue related to employee mobility and its consequences in their private sphere. Difficult conditions of the modern economy (both at the global and individual countries level - as here: the USA) require geographical changes of workplaces in order to ensure or maintain an appropriate level of financial security for individuals and their families. The purpose of the study was "to explore how employees who have relocated for employment opportunities perceive the costs and benefits of video technology as an option for maintaining family relationships for employees working in a company located in the United States". The implementation of such a goal made it possible to contribute to the area of social perception of employee relocation by identifying factors influencing the maintenance of relations with loved ones. The strengths of the article include: it deals with a current and important problem in the field of social sciences, it has a properly formed structure, a clearly indicated research gap, a well-defined goal and position in contemporary literature, aptly selected research methods with justification for the choice and a thorough discussion.
General concept comments:
This article has a logically ordered structure and all the necessary components seem to be included in it. The manuscript is clear and appropriate for the field of social science.
Literature analysis is carried out correctly. It is worth noting that the indicated research problem has been discussed extensively. First of all, the issue of employee relocation was discussed, second of all, the use of technology allowing for remote contact was characterized, and finally, it was noted that there are many studies that allow conclusions about the impact of using this technology on employees in the organization. In the background of these theoretical considerations, a research gap has been identified. The cited literature positions are recent. (1) It would be beneficial for the substantive value of the article to deepen the analysis of the literature by indicating additional positions contributing to the discussed topic.
The research methodology is described correctly. The choice of research methods was well justified, as was the selection of the research sample (including characteristics of the population and description of the selection of individuals for the sample). The procedure of the research, the contact channels with respondents, as well as the methods of collecting, coding and recording data were described in detail. This part of the article is solidly supported by references to the literature and the research choices are reliably justified. (2) To increase its value, it seems advisable to present the research tools in more detail - either by attaching them to the article or by describing them precisely.
Research results are presented in well-structured manner. The demographic profile of the participants was described, and then the results of the collected research material were presented in a condensed way. The strong point is the organization of considerations according to the research questions posed. The discussion as well as the recommendations for practice and the limitations of the research appear to be correct.
The literature referenced in this article is recent and the table is easy to understand. The presented research material allows to draw the presented conclusions.
Author Response
I am submitting a revised version of my original case study entitled “Relocated Employees’ Experience With the Costs and Benefits of Video Technology for Maintaining Relationships”. I have addressed the feedback provided as follows.
The first comment for revision was (1) It would be beneficial for the substantive value of the article to deepen the analysis of the literature by indicating additional positions contributing to the discussed topic.
In order to address this feedback, I have revised the review of literature, adding more material and organizing it into four sub-sections: Employee Relocation and Family, Technology for Video Communication, Benefits and Costs, and Technology and Communication with Family. This discussion extends from page 1 to page 3.
The second comment for revision was (2) To increase its value, it seems advisable to present the research tools in more detail - either by attaching them to the article or by describing them precisely.
I have added more details and sample questions for the data collection sources. This can be found on pages 5 and 6 in the revised manuscript.
Thank you for your consideration of this manuscript.
Reviewer 2 Report
1. 24% plagiarism found as per the turn it in plagiarism software. Authors are required to give the proper citations for the work that has been used in the article.
2.Proper formatting required in the whole paper especially in the reference section.
Author Response
Thank you for your feedback. I have ensured citations are accurately made and reference section is correctly formatted. This revision
extends from page 13 to page 15.
Reviewer 3 Report
PFA the document enclosed

Author Response
The first comment from Reviewer 3 was: The introduction is well developed. I suggest that the author/authors create a subsection on literature review. This section is important as it gives credence to the work already done, pinpoints the scarcity of extant work, and more importantly helps develop research gaps and provide a rationale to the study. Hence, the introduction may be crisp and relevant details may be taken up in the section on literature review.
I have addressed this by adding a literature review in the Introduction with four sub-sections: Employee Relocation and Family, Technology for Video Communication, Benefits and Costs, and Technology and Communication with Family. This discussion extends from page 1 to page 3.
The second comment from Reviewer 3 was: In the methodology section, the author/authors need to specify as to how they maintained ethics in the data collection process. What frameworks were utilized to maintain reliability and validity of the study? As NVivo has been used for thematic analysis author/authors need to specify the version too. Moreover, it is unclear as to why did they conduct both interviews and FGD in the same study. The purpose of these data collection mechanisms needs to be elaborated.
I have addressed this by adding more details on ethical measures and trustworthiness measures.
The third comment from Reviewer 3 was: The major themes drawn can read better if their description is interspersed with direct verbatim quotes from the informants themselves.
I have ensured quotes are present in the presentation of each theme.
The fourth comment from Reviewer 3 was: The section on discussion of results can be made crisp and succinct with details of address of research questions through the data analyzed. The following section on recommendations can be further subdivided into three subsections-Recommendations to theory,
Recommendations to Practice and Recommendations for further research.
I have shortened the section on discussion. There are no recommendations for theory.
The fifth comment from Reviewer 3 was: One of the most important sections of a manuscript is the implications of the study for theory and practice. Recommendations for theory are enriched when there is a proper review of literature at the beginning of the manuscript. Intuitive and counter intuitive
findings are then discussed in the light of the literature reviewed
.
I have addressed this by adding a literature review in the Introduction with four sub-sections: Employee Relocation and Family, Technology for Video Communication, Benefits and Costs, and Technology and Communication with Family. The discussion section draws upon this literature review.
The sixth comment from Reviewer 3 was: Similarly, recommendations for practice as presented in the manuscript are generic and need to be specific to generate value.
I have ensured the recommendations for practice follow the findings and do not extend beyond them.
Thank you for your review of this manuscript!
Round 2
Reviewer 2 Report
Dear Author ,
With the addition of the references and proper citations, you have the improved the quality of your research work.
Paper is ok now in terms of content clarity as well as originality.